# Fluid-Based Protein Biomarkers in Traumatic Brain Injury: The View from the Bedside

**DOI:** 10.3390/ijms242216267

**Published:** 2023-11-13

**Authors:** Denes V. Agoston, Adel Helmy

**Affiliations:** 1Department of Anatomy, Physiology and Genetic, School of Medicine, Uniformed Services University, Bethesda, MD 20814, USA; 2Division of Neurosurgery, Department of Clinical Neurosciences, University of Cambridge, Addenbrooke’s Hospital, Cambridge CB2 0QQ, UK; adelhelmy@doctors.net.uk

**Keywords:** brain, injury, phenotypes, biomarkers, mechanistic, clinical, utility

## Abstract

There has been an explosion of research into biofluid (blood, cerebrospinal fluid, CSF)-based protein biomarkers in traumatic brain injury (TBI) over the past decade. The availability of very large datasets, such as CENTRE-TBI and TRACK-TBI, allows for correlation of blood- and CSF-based molecular (protein), radiological (structural) and clinical (physiological) marker data to adverse clinical outcomes. The quality of a given biomarker has often been framed in relation to the predictive power on the outcome quantified from the area under the Receiver Operating Characteristic (ROC) curve. However, this does not in itself provide *clinical* utility but reflects a statistical association in any given population between one or more variables and clinical outcome. It is not currently established how to incorporate and integrate biofluid-based biomarker data into patient management because there is no standardized role for such data in clinical decision making. We review the current status of biomarker research and discuss how we can integrate existing markers into current clinical practice and what additional biomarkers do we need to improve diagnoses and to guide therapy and to assess treatment efficacy. Furthermore, we argue for employing machine learning (ML) capabilities to integrate the protein biomarker data with other established, routinely used clinical diagnostic tools, to provide the clinician with actionable information to guide medical intervention.

## 1. Introduction

Current studies involving blood-based traumatic brain injury (TBI) biomarkers have focused on correlations between a biofluid—typically blood/serum or plasma—level of a given biomarker in a select patient population and its predictive, or rather correlative, power based on the area under the Receiver Operating Characteristic (ROC) curve for TBI outcomes [1,2,3]. Such data only reflect a statistical association between a patient population, biomarker x and clinical outcome but do not in themselves demonstrate clinical utility. In order to utilize protein biomarker data to meaningfully contribute to clinical decision making, several criteria need to be met. These include (1) clear diagnostic value, i.e., the brain is indeed injured, and if so, then the type and extent of the primary injury in terms of cellular and molecular damage, and the identity of structures, i.e., vasculature, axons, etc.; (2) the immediate pathophysiological effects to the primary injury, e.g., excitotoxicity and other metabolic abnormalities; (3) the pathobiological components of the secondary injury process, e.g., inflammation; and (4) the temporal pattern of changes in biomarker levels as they relate to disease progression or regression as defined with other clinical outcome measures. These are tall orders, but all the necessary components are available: several potential mechanistic fluid-based protein biomarkers have already been identified [3]. Moreover, all the required technologies are readily available including analytical platforms enabling sensitive and multiplexed assays [4,5,6], as well as the necessary algorithms for data processing and machine learning (ML) [7]. However, to help to incorporate blood-based protein biomarker data into a patient management tool, two critical dimensions of TBI need to be considered, the spectrum of disease severity and complexity and the time factor.


The first dimension: The spectrum of traumatic brain injuries.


A key aspect of TBI pathology that could be supplemented with biomarker research is by reframing how TBI is defined. Traditionally, TBI is used as a single diagnostic term related to a mechanical insult to the head affecting the brain [8]. But TBI is not a disease itself and is rather the instigator of damage to the brain with consequent neurobehavioral abnormalities [9]. The only similarity between an unconscious patient with skull fracture, subdural hematoma and brain contusion and a patient walking into an emergency room (ER) with a bump on their head feeling dizzy is that there was a physical impact—of different kinds and intensities—to the head. Mild, complicated mild and repeated mild, moderate, and severe TBI are not a single disease but varying pathobiological consequences triggered with mechanical forces of different kinds and intensities [10,11,12,13,14]. TBI has been historically classified as mild, moderate, and severe according to the Glasgow Coma Scale (GCS) [15,16,17,18,19]. However, our biological understanding of the pathophysiological mechanisms underlying the functional abnormalities has become substantially more refined since the introduction of GCS [20,21,22]. For example, cerebral metabolic failure may be identified across the full spectrum of TBI severity and as such potential interventions for this abnormality require a robust and reliable indicator of whether a potentially tractable therapeutic target is present [23,24,25,26]. Conversely, there are examples of severe TBI where metabolic parameters are normalized while alternative mechanisms, such as a potent inflammatory reaction, may be at play [27,28,29]. These mechanisms are not mutually exclusive, and in reality, they overlap to greater or lesser degrees in a heterogenous patient population. The assumption that a single intervention randomized in a notoriously heterogenous disease can demonstrate a consistent effect is understandably undermined with the 100% failure rate of clinical trials [30,31,32,33]. It should be noted that there are signs of changing views in clinical TBI research [34]. The alternative approach to GCS-based classification is to define biomarkers that relate not to ultimate functional outcome, and can thus be confounded by the severity of injury, but to specific ongoing pathological process(es) [11,35,36,37,38,39]. This accords much more closely with potential therapies that can be translated from animal literature [40,41,42,43].

The physical injury, its type and its intensity determine the nature and severity of the primary structural damage, which then determines the secondary injury process aimed to minimize the extent of damage and to restore structural and functional homeostasis [44,45,46,47]. When combined, the primary and secondary injury, direct mechanical damage and the endogenous response to injury can cause complex pathobiological changes ranging from transient metabolic disturbance to massive cell death and tissue loss [48,49,50]. At the bedside, these pathobiologies manifest in varying degrees of neurological dysfunctions [51,52,53]. Severe TBI, frequently co-morbid with polytrauma, causes major loss of brain parenchyma, severe disruption of neuronal networks manifested in a coma and severe neurological dysfunctionality [54,55]. The pathobiological changes are the most complex after severe, especially penetrating TBI [56,57]. In contrast, there are comparatively fewer pathobiological changes after a mild TBI [58,59,60] (Table 1).

After moderate TBI, biomechanical forces can cause substantial direct tissue and cell damage and cell death, significantly disrupting neuronal signaling and networks, manifesting clinically as a prolonged loss of consciousness [34,61,62,63]. In the case of severe and moderate TBI, the various pathobiological changes occur in a partly overlapping fashion but the intensities and temporal pattern of the individual process differ and these processes interact in a highly complex fashion, such that one may trigger another [46,64,65]. Mild TBI on the other hand only causes temporary perturbance of cellular structures, and may dislocate membrane-bound ion channels, receptors and/or intracellular organelles, causing typically transient molecular disturbances reflected in metabolic abnormalities that clinically manifest as a temporary altered state of consciousness [34,51,66,67,68]. Complicated mild TBI, a subcategory that can be clinically defined as prolonged symptoms or an absence of full recovery [69,70,71], is caused by more severe mechanical damage to cells, and protracted disruption of neuronal signaling due to network disruptions [69,72].

The predominant pathobiological response to TBI-induced structural damage is inflammation [73,74,75,76]. The various forms and phases of the inflammatory process can be identified with a serial analysis of blood or CSF [77,78,79]. The inflammatory process starts immediately after the damage and its course may be one of the key determinants of the outcome after moderate to severe TBI [80,81,82]. Thus, identifying the components and temporal pattern of the inflammatory process can offer the acute care physician guidance for critical decision making about the need and type of medical intervention [80,83,84].

The second dimension: Temporal aspect of TBI-induced pathobiological changes.

A critical but currently poorly understood, and understudied, dimension of TBI is the temporal aspect of the overall injury process and the time-dependent changes of the associated pathobiological changes [44,85,86] (see Figure 1). Depending on the type and intensity of the physical forces, the primary injury is instantaneous and can vary from membrane perturbances, causing transient malfunction of ion channels, transporters and intracellular trafficking manifested as metabolic abnormalities, to massive tissue damage and bleeding [87,88,89,90,91,92]. The secondary injury process includes pathobiological responses to injury aimed at limiting damage and restoring homeostasis over a span of weeks or months [14,93,94,95].

The pathobiological responses can, at least partially, overlap and interact with one another [65,96,97,98]. Understanding the temporal aspect of the various pathobiological changes is important for physicians to make an informed decision about the need, type and, importantly, the timing of medical intervention. Biofluid-based protein biomarkers in TBI could assist, complement, and enhance the information content obtained from current, routinely used imaging, physiological and neurobehavioral/functional outcome measures [44,65,86,93,99,100]. They should accurately distinguish disturbance from damage, provide molecular-level information of the extent of damage and the pathobiological processes to guide evidence-based treatments, monitor treatment efficacy and assess the trajectory of the disease process.

In summary, serially obtained biofluid-based mechanistic protein biomarker data can—in combination with data of neural damage—greatly help to identify the disease processes and their temporal pattern. Such enriched protein biomarker data can be integrated with other—e.g., imaging—biomarkers to be part of the clinical decision-making process [101,102,103,104].

## 2. Protein Biomarkers in Clinical TBI; Current Status and Unmet Needs

Stressed, injured and dying neurons and astroglia release intracellular soluble molecules such as UCH-L1, NSE and S100B; structural proteins, e.g., GFAP, NF-L and tau, and these “neural damage markers” are the most utilized ones today and dominate publications [3,105,106,107,108,109,110,111]. They provide essential information about the severity of the primary injury, which is reflected in the extent of their blood or CSF levels. When they are assayed longitudinally, their chronically elevated levels are indicators of ongoing neural damage [112]. While indicating ongoing pathological processes that keep damaging or killing neural cells, these markers do not identify the pathobiological processes themselves that cause the ongoing parenchymal damage. In short, they cannot identify the disease (endo)phenotype. This is a huge, clinically important unmet need: one would not consider referring to SARS-CoV-2 only as an undifferentiated “infection” [113]. Without the precise understanding of the identity and temporal pattern of injury-induced pathological response, no evidence-based, specific therapy can be identified and delivered within a defined therapeutic window.

Increasingly, the importance of vascular involvement in the pathobiology of TBI is being recognized [35,114,115]. The extensive vascularization of the brain makes it especially vulnerable to biomechanical forces causing damage to endothelial tight junctions reflected in the release of tight junction proteins, e.g., claudin-5 [116] and/or occludin [117,118,119,120,121]. Other injury-induced vascular abnormalities such as endothelial stress or altered hemodynamics are reflected in elevated biofluid levels of VEGF (vascular endothelial growth factor), vWF (von Willebrand Factor), ADAMTS13 (a disintegrin and metalloproteinase with a thrombospondin type 1 motif, member 13), etc. [114,115]. The forces can also dislodge cells and disrupt cell–cell adhesion as indicated with elevated blood and CSF levels of VCAM-1, CNX43 or MMP9 [122,123,124,125,126,127]. Importantly, some of these proteins, e.g., CNX43 and MMP9, are activators and or mediators of the inflammatory response to the insult [27,29,73,74,79]. Damaged and/or dying cells release damage-associated molecular patterns (DAMPs) and intracellular molecules, like S100A8/9, hsp70 and HMGB1, that collectively activate the innate immune system’s response to injury [128]. This process, if uncontrolled, can develop into a chronic inflammatory process [80,82,83,84]. Circulating inflammatory molecules represent the group of fluid-based protein biomarkers that can be called “mechanistic markers”, i.e., markers of specific pathobiological processes [1,129]. IL-8, activated by MMP9, is a proinflammatory and angiogenic signaling molecule; CXCL12, activated by HMGB1, is an inflammatory and angiogenic signaling molecule. Thus, CXCL12 and other inflammatory molecules connect damage to repair mechanisms of which inflammation is a critical component [130].

There is therefore a distinction between the non-specific markers of cell injury and death and those that may reflect a specific pathological process. In the case of the first category (e.g., UCH-L1, GFAP), the greatest value is in determining the cumulative burden of injury either to exclude significant TBI at the mild end of the spectrum, or to prognosticate at the severe end of the spectrum. Between these extremes, confounders related to blood–brain barrier permeability, contribution from extracranial sources and timing in relation to injury make interpretation in relation to a defined threshold more difficult.

In the second category of markers that relate to a specific pathobiological process, they may reflect the mode of the ongoing injury process rather than the magnitude. It is this category that has been relatively under-investigated and but will have roles in patient stratification (‘phenotyping’) to deliver specific therapies. In all likelihood, the time-dependent changes in the biofluid levels of these two categories of protein biomarkers will be in a clinical Decision Tree [11,131,132,133]. The clinical needs and status of currently used fluid-based protein biomarkers using the examples of neural injury markers are summarized in Table 2 and examples of the sensitivity and specificity of the selected legacy protein TBI markers are listed in Table 3. While the list of desirable factors seems very stringent or potentially impossible to achieve, in relation to protein biomarkers, however, if one were to consider the CT head for post traumatic hemorrhage [134,135], it would fulfil all of them.

## 3. Biofluids Available for Protein Biomarker Analysis in the Clinical Setting

The wide spectrum of clinical presentations in TBI determines which biofluids are available for a protein biomarker analysis (Table 4) [13]. In mild TBI, blood and plasma/serum are the most commonly used, although saliva has received some interest albeit for microRNA-based investigations [150]. In moderate and severe TBI, the availability of invasive monitoring and repeated sampling provides an opportunity for a different panel of biofluids as well as the opportunity to consider trajectories of change and AUC approaches [46]. CSF sampling is typically via an external ventricular drain (EVD) in the context of moderate and severe TBI, given the presence of raised intracranial pressure (ICP) and the risk of cerebral herniation with sampling of CSF from the lumbar cistern. In some units, EVDs are used as both a means of monitoring ICP as well as a method for controlling raised ICP with intermittent or free drainage. With free drainage, there is often a situation in which raised ICP leads to collapse of the ventricular ependyma around the EVD, temporarily occluding the EVD, and making sampling difficult. There is an additional concern about infection, particularly with prolonged use of EVD and with repeated sampling and flushing. CSF acts as a sump for fluid traversing the brain parenchyma and, as such, is a global sample of brain-produced proteins. As the volume of CSF drainage tends to be in milliliters per hour, there is often no limit on the analytical methods available to be used [34,54,55].

MD sampling is unique in that it samples the brain extracellular fluid in a continuous fashion [151,152,153,154,155,156,157,158]. As such, it can sample a biological compartment in which several proteins act on cell membrane receptors. There are qualitative and quantitative differences in the concentrations of biomarkers between CSF and MD sampling that are likely to reflect genuine differences in the biology of the two compartments. However, MD sampling is a specialized technique, which is not universally available outside specialist intensive care units. MD relies on the diffusion of substances in the brain extracellular space across a semipermeable membrane in the microdialysis catheter where a carrier fluid is pumped continuously and subsequently collected and analyzed [151,159,160]. Each molecule crosses the microdialysis membrane at a different rate, depending on the molecular weight, hydrophilicity, oligomerization and membrane porosity. The fractional concentration that crosses the microdialysis membrane is termed the relative recovery and is influenced by the specific MD methodology employed. This requires careful control of the type of MD catheter used (‘Molecular Weight Cut Off’), the type of perfusion fluid and fluid perfusion pump rate. When considering an MD biomarker, these factors need to be consistently reported and utilized.

MD is a focal monitor that samples a region of the brain that is constrained by the diffusion distance immediately around the microdialysis catheter. This can be useful when MD catheters are placed adjacent to a focal lesion, or when an abnormality is so widespread that it can be detected in any portion of brain. However, in some circumstances, the region of the brain that is monitored is not representative of the entire brain and is therefore not useful for guiding treatment.

The argument for CSF and microdialysis sampling therefore depends on identifying a clear biological rationale for the additional inconvenience and patient risk associated with additional monitoring [65]. Taken together, sampling of biological fluids from the central nervous system compartment has two clear advantages. Firstly, in the context of trauma, there may be extra-cranial release of biomarkers, depending on the CNS selectivity of the biomarker in question. Secondly, the release of CNS proteins into the peripheral circulation requires passage across the blood–brain barrier, which may be variably disrupted (in time and location across the brain cerebrovascular bed), such that an additional unmeasured factor impacts on the variation of a measured biomarker [13]. Although CSF and microdialysis sampling have been used widely in clinical research literature, blood and plasma remain the most clinically accessible and acceptable biofluids [13]. Importantly, however, the quantitative and temporal relationships between biomarker levels in matching biofluid blood, CSF and cerebral microdialyzates are currently not well established [64,65].

## 4. The Role of Protein Biomarkers in the Clinical Decision-Making Process for Various TBI Severities

There are several factors that should guide integration of protein biomarker data into the clinical decision-making process after TBI. As outlined above, there is a spectrum of severities and conditions induced with TBI. Protein biomarkers can provide a varyingly critical role in the clinical decision-making process depending on the severity—and type—of injury (see Table 5 and Table 6). In case of a severe TBI with skull and dura penetration and cranial bleeding, a diagnosis that the brain is injured can be made without protein biomarkers of neural damage. On the other end of the TBI spectrum, measuring blood levels of markers of neural damage can provide important diagnostic information—not obtainable with other tools—in determining whether the confused or semi-conscious individual who is being brought into the ER without any sign of physical impact has suffered any physical brain damage or not.

## 5. Mild TBI and Concussions

Despite the classification, mild TBI (mTBI) is a very challenging condition from the clinicians’ perspective [161]. A concussion or mTBI has the highest prevalence and incidence among TBI cases and the clinical symptoms can be similar to intoxication, poisoning, drug use/overdose or metabolic crises [162,163].

In addition, the time of injury in relation to clinical presentation and assessment can be delayed by several days as individuals may seek medical help at different post-injury time points and the exact time of the impact is frequently not known. The three most affected age groups, infants, young adults and the geriatric population, have very different neurodevelopmental status, biological backgrounds, biological reserve capacities, co-morbidities and medications. Thus, diagnosing that the brain is indeed physically injured is as important as it is challenging (Table 5).
ijms-24-16267-t005_Table 5Table 5Biofluids for protein biomarker analysis after mild TBI relevant for the clinical decision process.BiofluidProsConsIssuesUnmet NeedsBlood (serum, plasma)Easy, minimally invasive, isolating serum/plasma well establishedIntracranial origin of mechanistic biomarkers is unclearBlood may be collected, and processed outside of clinical lab setting, affecting quality; cell lysis can occur during clothing contaminating serum with intracellular components forming white blood cells; the choice of anti-coagulant can affect assay; platelets can contaminateQuality control of the input biofluid (plasma and/or serum) for intactness; reference ranges for normal values [164]ExosomesPotential to improve brain specificityLengthy, not standardized isolation; requires ultra-sensitive and lengthy assaysMost studies use frozen blood as source, purity and brain specificity are issuesWell-established, easy, standardized isolation procedure; quality control; reference ranges for normal values

The current Duoset, UCH-L1 and GFAP, by Abbott was initially authorized to assess the need for CT scans after head impacts, assuming that a positive UCH-L1/GFAP readout indicates structural damage that might require intervention [165,166,167]. However, the specificity of Duoset detecting a concussion has been challenged and so far, there has not been indication of a decreasing use of CT scans in ER settings [168,169]. This is understandable because the two modalities provide very different clinical information that can only partly overlap or substitute for one another. A CT scan is to determine if the head-impacted individual has intracranial hemorrhage—either subdural, epidural, intracerebral or subarachnoid—cerebral edema and/or skull fracture requiring surgical interventions. A positive CT scan virtually always correlates with elevated UCH-L1/GFAP levels but not the other way around as elevated UCH-L1/GFAP levels may or may not indicate structural damage detectable with imaging. However, current blood-based protein biomarkers of neural damage, Abbott’s Duoset and Quanterix’s fourplex (NF-L, tau, GFAP, UCH-L1), can play an important role in concussion diagnoses, assessing the rate of recovery and the efficacy of disease management. Importantly, these tests are not expected to replace CT scans as several studies have shown elevated blood levels of these markers in other CNS disorders like Alzheimer’s Disease [111,170] and also in non-neurological disorders [171,172] including COVID-19 [84].

One great success in this field has been the incorporation of the S100b assay in the emergency department guidelines for assessment of mild or minimal TBI in the modified Scandinavian Neurotrauma Committee Guidelines [173]. A single measurement of S100b within 6 h of injury allows 20% of low-risk patients presenting with a mild or minimal TBI to be discharged without a CT scan or observation [174]. While this may seem a modest percentage of patients, the large bulk of patients presenting to emergency departments are at the mild or minimal end of the spectrum so in health economic terms, this is a significant achievement [175]. Even where a plasma biomarker has been incorporated into a clinical guideline, several practical limitations exist. Overall compliance to the guideline is 40% with 50% of patients with S100b below the designated threshold having a CT scan irrespectively. In this cohort, the S100b assay has not contributed to the clinical management and may lead to delays to patient management. Typically, there is a circa 2 h delay in securing an S100b result and in those patients where a negative result is ignored, or a positive result mandates a CT scan, then there is a delay in organizing a scan following the S100b assay, leading to a delay in discharge as compared to a CT scan as the first point of assessment.

The mTBI or concussion cohort is an appealing group of patients in order to utilize plasma biomarkers as with a sufficiently low threshold, a high sensitivity can be achieved, maintaining a high level of safety and clinical confidence. There is a theoretical risk that an unrecognized fracture, without any underlying brain injury, causing a putative increase in S100b, could cause subsequent deterioration from an extradural hematoma. Nevertheless, taken together with appropriate advice to the patient, the chances of missing a clinically significant injury are very low. The difficulty for all biomarkers in the acute setting when compared with a CT scan is that a CT scan has near 100% sensitivity and specificity for clinically significant brain injury. With the ubiquity of this scanning modality in the developed world, the performance of any potential plasma biomarker has a very high threshold to achieve clinical utility. Within the Scandinavian Neurotrauma Guidelines, the argument has been made that S100b is a ‘radiation-sparing’ adjunct rather than a method of supplanting a CT scan in the assessment of mild or minimal TBI. There is interest in alternative biomarkers such as UCH-L1 and GFAP as potential biomarkers that can reduce the reliance on a CT scan with very high sensitivity and specificity for CT-negative TBI [139]. This would suggest that the specific CNS protein biomarker that is used is not critical but there is a general premise that proteins, which should be constrained by the blood–brain barrier, leak into the blood following trauma, either because of a leaky blood–brain barrier; damage of CNS cells, leading to an increase in CNS levels of these proteins; or most likely, both. The high predictive power of these biomarkers does not therefore imply a mechanistic role for the specific proteins in the pathophysiology of TBI, notwithstanding that S100b and GFAP are predominantly astrocytic while UCH-L1 is neuronal. A PCA analysis of a panel of biomarkers demonstrates the high co-linearity between several of them [39], suggesting that they contain overlapping information in relation to the variability between patients.

## 6. Moderate/Severe TBI

In moderate/severe TBI, the argument for clinically relevant biomarkers—currently neural damage markers—is more complex [46]. In this circumstance, these biomarkers are not relevant in making a diagnosis as the clinical picture and CT findings contain the most clinically relevant information (Table 6). The ‘prognostic’ powers of these damage biomarkers or existing multivariate models such as CRASH and IMPACT are undoubtedly important research tools that can be successfully utilized to stratify patient groups. However, from a clinical perspective, they have limited utility for several reasons.
ijms-24-16267-t006_Table 6Table 6Biofluids for protein biomarker analysis after moderate and severe TBI relevant for the clinical decision process.BiofluidProsConsIssuesUnmet NeedsBlood (serum, plasma)Easy, minimally invasive, isolating plasma and serum well establishedExcept for the true damage markers (USCH-L1; GFAP; tau; NF-L), the intracranial origin of mechanistic biomarkers is unclearCell lysis can occur during clothing contaminating serum with intracellular components forming white blood cells; the choice of anti-coagulant can affect assay; platelets can contaminateQuality control of plasma and/or serum for integrity; normal ranges not standardizedCSFReflects intracranial fluid milieu; closeness to brain parenchymaLacks region specificity, low global damage and high level of focal damage can result in the same biomarker levels Potential blood contamination reduces diagnostic valueQuality control for CSF integrity; normal ranges not standardizedbEDF/cMDReflects intraparenchymal changes Highly regional, very low volume, requires high-sensitivity assays Limited number of clinical sites use it. Recovery affected by the size of proteins, chargesQuality control for CSF integrity; normal ranges not standardizedNotes: The fluid dynamics and molecular movements between the different compartments are currently poorly understood [176].

Firstly, even if these biomarkers were to predict a poor outcome, it is rare to withdraw therapy altogether based on this prediction. In practice, a multitude of factors are considered of which the clinical assessment of the patient, predominantly CT scan findings, GCS and pupillary function, after appropriate resuscitation and control of intracranial pressure is key. If the biomarker agrees with the clinical assessment and CT findings, it does not add useful information. If it provides a contrasting picture to clinical assessment and CT findings, then it would not take primacy in decision making. While it has been shown that, for example, GFAP concentrations are effective predictors of CT abnormalities [177,178] in the clinical arena, one would not envisage GFAP *replacing* a CT scan and therefore the key question is how GFAP results *add* prognostic information to CT findings that would impact on treatment. This is not clearly evaluated in the current literature.

Secondly, many, if not most, biomarker studies employ assays of biomarkers in a limited time frame, typically only on admission. From a methodological standpoint, it is wise to limit the intrinsic heterogeneity that occurs when sampling occurs at different time points, in what is already a heterogenous disease. However, this approach takes no account of the response to treatment or intervention. In the clinical arena, the trajectory of improvement or deterioration contains important decision-making information in relation to the prognosis. When patient’s families are advised with regards to the prognosis, it is normal practice to be guarded to reflect the significant heterogeneity of the functional outcome after TBI. This makes the ‘average’ prognosis with a given combination of variables or conditions on admission of limited value. Experienced clinicians will commonly encounter surprising outcomes, both much better and much worse than would have been predicted solely on admission parameters. The nature of TBI at the severe end of the spectrum is such that there are often several opportunities to limit the ceiling of care or even consider withdrawal of therapy. It is therefore unnecessary to make a definite decision on a poor prognosis on the bases of admission parameters. There have been advances in modelling the time course of biomarkers following TBI, e.g., Ercole et al. [179]. This approach could be used to identify deviations from this time course as a metric of additional injury. However, there is significant additional complexity in incorporating this into a clinically applicable tool [164]. Thirdly, from the utilitarian clinical perspective, one must consider the therapeutic interventions that can be introduced on the basis of additional information provided with a biomarker. The bulk of management strategies directed at TBI are supportive and targeted to maintaining physiological parameters within a range, which is perceived to be supportive of neuronal function and therefore minimizes secondary injury. As such, every patient is treated in a very similar way, largely framed in relation to blood pressure/cerebral perfusion pressure, systemic oxygenation/brain tissue oxygenation and intracranial pressure. Multiple additional monitoring techniques are employed in differing units and contexts; however, none of these interventions address the specific information provided with a protein biomarker. If no additional therapy can be introduced because no biomarker-specific therapies exist, then there is no way that the additional information can impact on patient care. ‘Early warning’ markers have been postulated as possibly preventing a deterioration before it happens; nevertheless, the therapeutic interventions we have available remain limited.

It is important to note that elevated blood levels of most of the currently used protein biomarkers can also be the result of injuries to organs other than the brain. However, studies have shown that injury to peripheral organs can also affect the TBI disease process. Within research settings, the utilization of polytrauma models and metrics (such as injury severity scores) can provide critical information about the contribution of peripheral injuries on the outcome of TBI, but most experimental studies do not include injuries to other organs. In the clinical domain, especially in an emergency department setting, multiple factors limit the ability to determine the contribution of peripheral trauma to the TBI disease process. Firstly, because additional injuries may be occult at the time of presentation, and secondly, because clinical implementation protocols are designed to be used by those who do not necessarily have specialist knowledge of the biomarker in question, and therefore may not appreciate the confounders introduced with extra-cranial production of a given biomarker. Additional complicating factors may include hepatic or renal clearance of a biomarker, which is impacted by patient-specific factors outside the index trauma. Systemic blood further averages and dilutes any CNS-specific changes. Furthermore, the dynamic relationships between these fluid compartments regarding molecular transport, clearance and degradation are currently poorly understood.

## 7. Issues with Integrating Protein Biomarkers in the Clinical Decision-Making Process to Improve Diagnosis and Prognosis and to Guide Treatment

Taken together, there remains a role for research in implementation of specific clinical protocols that incorporate protein biomarkers. These will require careful consideration of the role of the biomarker in the decision-making algorithm and consideration of the relevant outcome metrics. For example, a reduction in the requirement for clinical follow up in a low-risk mild TBI group may reduce resource utilization and burden on healthcare institutions without necessarily having any impact on clinical outcome. These approaches are distinct from the role of biomarkers as a read-out of the underlying pathology following TBI of all severities and types.

Having recognized that in the narrow clinical setting, there are many limitations of protein biomarkers, it is worth considering what an ideal TBI biomarker might be or what it could or should achieve. The heterogeneity of severity, pattern of injury, pathophysiological mechanisms and their temporal pattern of changes along with the outcomes make TBI a difficult condition to study and the premise that a single biomarker can encapsulate the complexity of the condition has limited the field. Rather than considering whether a biomarker can predict a CT scan abnormality or prognosis, a more fruitful approach would be to stratify the underlying pathophysiological mechanisms that drive TBI-induced injury, using a whole suite of biomarkers.

This will require a different panel of biomarkers than has previously been considered as the focus has previously been on cellular specificity of the relevant proteins (e.g., neuronal vs. astrocytic). In this way, the finding that a biomarker correlates with the outcome is not in itself important. The key issue is whether the level of a given biomarker reflecting a distinct pathology is at play. In this way, rather than framing TBI as a single pathology, biomarkers can stratify patients regarding the underlying pathophysiological insults (metabolic, innate inflammation, autoimmune) in an analogous way to the classification of injury patterns (contusion, extra-axial hematoma, diffuse axonal injury) [10]. In other words, using protein biomarker signatures to identify the disease (endo)phenotype [35,38]. Moreover, the clinical utility of protein biomarkers is also dependent on the severity and the extent of functional incapacity of TBI.

In the context of moderate-severe TBI, there is therefore a need to extend beyond the standard biomarkers of neuronal and astrocytic injury and consider a wider spectrum of mediators. A range of pathological mechanisms occur following TBI, and specific inflammatory mediators may provide discrete information that addresses the mechanistic link between the mechanical disruption that occurs at the time of injury and how this translates into neuronal and astrocytic cell death. This can include Damage-Associated Molecular Patterns (DAMPs), such as HMGB1, free DNA or mitochondrial DNA [128]; markers of innate inflammation such as interleukin (IL)-1, IL-6 or IL-10 [180,181,182]; markers of mitochondrial dysfunction [183]; or markers of adaptive immunity such as autoantibodies to neural epitopes [80,81]. Taken together, there is a wealth of potential information from these biomarkers that, in principle, can guide targeted therapies. However, current understanding of the complexity of these responses is still embryonic and there is a dearth of potential therapeutic avenues that limit clinical utility.

Patients with moderate and severe TBI are in a high-dependency setting and therefore there is the opportunity to carry out sequential sampling to build up a trajectory of response. Interpretation of this trajectory is complicated with the natural history of biomarker release, which may depend on several factors such as burden of injury, change in blood–brain barrier permeability, hepatic or renal metabolism, therapeutic intervention or hypothermia for example. Thus, interpretation would require a robust understanding of the expected evolution of biomarker levels in order to determine whether the levels in any given patient were higher or lower ‘than expected’. This longitudinal trajectory could then be used to assess the success of intervention or additional burden of secondary injury. Currently, this conceptual framework is still an aspiration as this approach has not been widely used in the literature. However, for biomarkers to be clinically useful, a large amount of detailed work needs to be carried out over and above ‘this biomarker correlates with outcome’.

One approach, with increasing interest in the literature, is a range of statistical approaches under the umbrella of Big Data Analytics [184] or Artificial Intelligence (AI) and machine learning [7,185,186]. There are a wide variety of mathematical models that can be utilized to try and identify occult relationships between variables using brute force statistical approaches from large aggregates of data. These methods have clear benefits in that they are unbiased and do not have to rely on any pre-supposed assumptions. The promise of uncovering hitherto unrecognized patterns in data is a seductive one. These algorithms’ output must be used *in toto* as it is very difficult or impossible to retrospectively analyze the resulting code to generate a meaningful biological interpretation. In a practical sense, an algorithm could be validated and used as a ‘black box’, but one would never know, in any one circumstance, which of the input variables in combination was responsible for the resulting output. The question then becomes whether the output is sufficiently useful for clinical practice to be worthwhile by changing patient management, and whether the clinician is confident enough to base decisions on this output when harm may result. Currently, these approaches are somewhat theoretical in relation to biomarker utility but are likely to become an increasing feature within the literature. Examples have shown that a Decision Tree can be built if there are enough “clean” clinical data, e.g., the disease outcome is well defined and—preferably longitudinally obtained—protein biomarker data are available [131,187]. Given the ultra-fast advancement in utilizing AI (see “Generative AI”), these approaches are feasible but they are currently hindered with the lack of high-quality standardized, reproducible unified protein biomarker data that are available in a high quantity. (See also below under Unmet Needs)

## 8. Unmet Needs

The field of protein biomarkers has become a key part of the landscape of TBI research in the pre-clinical and clinical domains. Alongside the technical aspects of assay development, there is a need to consider implementation methodologies that incorporate the key recommendations we have set out in this regard. The unparalleled progress in TBI biomarker research has still not resulted in clinical utility. Below is a list of a few issues that need to be addressed toward that goal. Some of them are purely technical, and others would require changing how we think about a protein biomarker in TBI, changing habits and practices.
(1)Standardization: The number one technical issue is a lack of standardized preanalytical and analytical procedures and clinical-grade standardized assay platforms. Studies have shown that preanalytical variables majorly affect the quality of the analyte—blood/plasm/serum, CSF, etc.—and consequently the output data. Preanalytical variables include the procedures of collecting, handling, processing, storage and transportation of biofluids. Combined, these factors can change the output data as much as an order of magnitude (!). These preanalytical variables must be of special concern for settings outside of a clinical lab—sport fields, military field environment—because patients with moderate and severe TBI are in a clinical setting with strict medical procedures [164]. In the absence of indicator(s) of sample quality, like the 28S-to-18S ratio for RNA work, employing strict preanalytical procedures outside of a clinical setting is essential toward establishing blood-based protein biomarkers. Measuring hemoglobin contamination is an important first step toward that goal. Protein degradation should especially be a serious concern because virtually all analytical platforms are antibody-based. Their accuracy (and specificity) relies on the intactness of epitope(s) specific for the protein biomarker of interest. Different antibody-based analytical systems utilize different antibodies, which are typically proprietary information. Not surprisingly, the biomarker values of identical input samples varied significantly between the different platforms used for an analysis [188]. Damaged or altered epitopes can especially be consequential for the high and/or ultra-high sensitivity analytical platforms. Mass spectroscopy can offer a potential solution but the current technology—while it is rapidly evolving—still cannot meet the sensitivity and speed required for clinical utility.(2)Specificity: Due to the cellular and molecular complexity of the brain—parenchyma and stroma—and the similarly complex pathobiological responses induced with TBI, there is no single biomarker—the “troponin of TBI” has emerged as having clinical utility. The six markers—UCH-L1, GFAP, S100B, tau/pTau, NF-L—are not only different in their cellular specificity and intracellular origin—e.g., soluble vs. cytoskeletal—but their biological half-lives, clearances from the intracranial space and their stability in the extracellular milieu are vastly different. None of them used individually are specific for TBI but elevated biofluid levels simply reflect neural cell damage that can be caused by other insults or pathologies. However, the specificity increases when they are co-analyzed as a biomarker panel. The use of such a “sixplex” will have improved specificity, accuracy and sensitivity and will reduce false negative protein biomarker results. Because these proteins are of different molecular entities, different stabilities and different biological half-lives, the challenge is not only to co-analyze them reliably and reproducibly in biofluids but to create an algorithm that takes their biological and molecular variables into account.(3)Reference ranges: Clinically utilized fluid-based biomarkers of metabolism, inflammation and organ damage—e.g., troponin—have well-established and verified normal reference ranges based on values found in healthy individuals. Moreover, some of the markers have established reference ranges for different age groups, biological males or females and races. No such reference ranges exist for any of the neural damage markers—GFAP, UCH-L1, NF-L, tau—and their various phosphorylated forms—used in most clinical studies. In the absence of such reference ranges, the current and published fluid biomarker data, which widely vary from laboratory to laboratory, have only limited, if any, clinical utility.(4)Longitudinal studies: It has been amply demonstrated that TBI-induced pathobiologies change over time, but most studies provide only single-timepoint-based protein biomarker data. Serial sampling and biomarker analyses are critical to identify the ongoing pathobiological changes, their onset and their extent. Such an approach will identify potential therapeutic targets, their therapeutic windows as well as disease trend and potential outcome. Such serial sampling and analyses, even if they are restricted to the “classic”, neural damage markers—GFAP, UCH-L1, NF-L, tau and its various phosphorylated forms—will provide clinically useful information about the disease trend. Elevated biofluid levels of these markers beyond the acute phase will indicate ongoing processes that keep damaging and/or killing neurons, glia and axons.(5)Expanded biomarker panel: Neural damage markers alone cannot identify the pathobiologies causing the extended damage. Measuring time-dependent changes in the biofluid levels of “mechanistic” protein biomarkers and markers of endothelial stress, vascular injury, cell adhesion, inflammation, etc., will identify the pathobiological changes and importantly their temporal pattern. If we know the pathobiologies, we can identify potential therapeutic targets and/or therapies. Determining the time-dependent changes in the biofluid level of protein biomarkers will help to identify potential therapeutic windows for pharmacological or other interventions. The good news is that several of such “mechanistic” markers have already been routinely analyzed in clinical/ER settings and can identify ongoing pathobiological processes, e.g., inflammation. The combination with neural damage markers such as an extended biomarker panel—if the technical, etc., issues listed above are addressed—can serve as the basis of clinical utility.(6)Biomarker data management and integration: Currently, biomarker data are deposited in medical records, and only a fraction of the data are available typically through scientific reports or publications [189]. The biomarker data in those reports or publications are unstructured and not readily available for data mining, machine learning (ML) or other Artificial Intelligence (AI) approaches. Depositing fluid-based protein biomarker data in a database, e.g., maintained by the NIH or insurance companies, will also allow to harmonize and integrate fluid-based protein biomarker data with imaging, physiology, neurological, etc., data currently collected in clinical settings and thus taking advantage of the ever-increasing power and capabilities of Generative AI [190].

## 9. Summary

Here, we have reviewed and discussed the status of fluid-based protein biomarkers in TBI and outlined what needs to be carried out to transition from a research tool to becoming embedded within a clinical paradigm (Table 7). It requires a change in focus in biomarker research toward clinical utility by establishing the criteria described in the ideal biomarker box. A second stream of work is then required to determine the specific protocol that would utilize the biomarker within a diagnostic and/or therapeutic pathway and establish the evidence base. The outcome of this pathway can be improved regarding efficiency of assessment or a health economic benefit outside the narrow remit of the patient outcome. This is a very different approach to those used in most biomarker research.

## Figures and Tables

**Figure 1 ijms-24-16267-f001:**
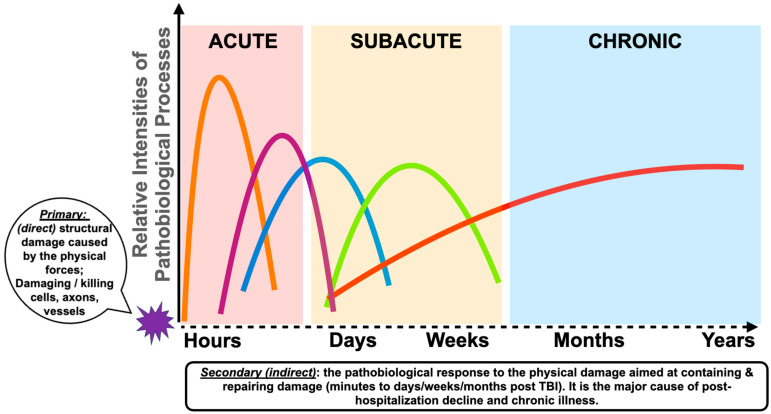
Hypothetical timeline and relative intensities of the pathobiological process after moderate to severe TBI. The colored lines indicate suspected pathobiological changes, metabolic (orange), axonal damage (purple), glia damage (blue), vascular abnormalities (green) and inflammation (red), during the acute, subacute, and chronic phase post injury.

**Table 1 ijms-24-16267-t001:** The presence and relative intensity of known pathobiological changes after various severities of traumatic brain injury.

TBI Severity Pathobiology/Abnormalities	Sub-Concussive	Mild/Concussion	Repeated/ Complicated Mild	Moderate	Severe
**Metabolic**					
- hypoxia	-/+(?)	-/+(?)	+(?)	++	+++
- oxidative stress	-/+(?)	-/+(?)	+(?)	+	+++
- cerebral glucose	-	-/+(?)	+(?)	+	+++
- excitotoxicity	-	-/+(?)	+(?)	+	+++
**Neuron, astroglia**					
- stress/damage	-	-/+(?)	+	++	+++
- loss/death	-	-	-/+	++	+++
**Axon (TAI, DAI)**					
- stress/damage	-	-/+(?)	+	++	+++
- loss	-	-	-/+	+	+++
**Vascular/endothelial**					
- stress	-/+	+	++	+++	+++
- damage	-	-/+(?)	+	++	+++
- (micro)bleeding	-	-	-/+	+	+++
**Cerebral edema**					
- cytotoxic	-	-	-	-	+/-
- vasogenic	-	-	-	+/-	++
**Inflammatory response**					
- neuroinflammation	-	?	-/+	++	+++
- autoimmune	-	-	?	+/-?	++?

**Legend:** - = not present/detected; -/+(?) = reported in some cases but not generally established; ? = unknown; + = present but not dominant; ++ = dominant; +++ = leading pathobiology. Yellow highlight = currently detected with physiological (e.g., jugular venous oxygen saturation (SjvO_2_)) monitoring, near-infrared spectroscopy (NIRS) and brain tissue partial pressure of oxygen (PbtO_2_), and/or imaging positron emission tomography (PET), near-infrared spectroscopy (NIRS) and magnetic resonance imaging (MRI); in severe TBI when bECF is collected by cMD biochemically; Orange = currently mostly detected with imaging, and protein biomarkers are available and have been used especially as neuron and astroglia damage markers: Green = protein biomarkers available but currently not in routine use. **Notes:** the current TBI injury severity scale is based on the extent of impairment of neurological functions assessed using GCS and using structural damage assessed with imaging. We currently have limited understanding of the relationship between the extent of functional impairments and the blood (and/or CSF) levels of protein biomarkers. Moreover, the “biological reserve/resilience”, age, comorbidities and comedications of the injured individual can greatly modify the diagnostic and predictive value of current molecular markers.

**Table 2 ijms-24-16267-t002:** Current status of most referenced biofluid-based protein biomarkers used in TBI *.

Technical Requirements	Can Be Measured in Easily Available Biofluids	Can Be Assayed Multiple Times	High Sensitivity, Specificity Assays Available	Standardized Assay Platform	Standardized Outputs	Rapid Results
Requirements Fulfilled	Yes	Yes	Yes (platform dependent)	No	No	No
Discrepancy	N/A	N/A	N/A	Critical	Critical	Critical
**Clinical** **Requirements**	**Normal Reference Ranges** **Available**	**Disease-Related Trajectory of Results Defined**	**Conceptual** **Understanding** **of Results**	**Results Identify Therapeutic** **Interventions**	**Results Help with Avoiding or Withdrawing Harmful Therapies**	**Results Reflect Success of Therapeutic Interventions**
Requirements Fulfilled	No	No	Yes	No	No	No
Extent of Discrepancy	Critical	Critical	N/A	Critical	Critical	Critical

**Note:** * = UCH-L1, GFAP, NSE, S100B, NF-L, Tau.

**Table 3 ijms-24-16267-t003:** Sensitivity and specificity of selected legacy protein biomarkers used in various forms of TBI *.

Biomarker	Abbr.	Sens.	Spec.	Notes and References
Ubiquitin C-Terminal Hydrolase-1	UCH-L1	0.97	0.40	Mild TBI; +/- Intracranial Lesions [136]
Glial Fibrillary Acidic Protein	GFAP	0.93	0.66	All Severities: +/- Intracranial Lesions [137]
		0.99–0.84	0.15–0.59	Mild TBI; CT+/-: Concentration Ranges of 13.1–190.1 pg/mL [138]
DuoSet	UCH-L1/GFAP	0.976	0.364	Mild TBI; CT+: Predetermined Cut-off of UCH-L1 = 327 pg/mL; GFAP = 22 pg/mL [139]
Neuron Specific Enolase	NSE	0.79	0.50	Severe TBI; Mortality [140]
		0.72	0.66	Severe TBI; Unfavorable Neurological Prognosis [141]
Calcium Binding Protein S100 Subunit Beta	S100B	0.95	0.47	Mild TBI; +/- Traumatic Intracranial Hemorrhages [141,142]
Neurofilament Light Chain	NF-L	0.72	0.96	Favorable Outcome (GOSE > 5) [143,144]
Tau Protein	Tau	0.88	0.94	Predicting Poor Outcomes [145]

**Abbreviations:** Abbr. = abbreviated name; Sens. = sensitivity; Spec. = specificity. * **Caveats**: There are no “generic” specificity and sensitivity of any of the biomarkers. Sensitivity and specificity of any given biomarker are influenced by many factors, some of them listed here: (1) Clinical variables: (a) severity of injury; (b) type of injury; (c) type of outcome measures (e.g., CT+ vs. CT- or dead vs. alive); (d) time elapsed since TBI and blood sampling for analysis. (2) Analytical variables: (a) type of assay platform used, (b) assay sensitivity, (c) analytical ranges and cut-offs used [146,147,148,149].

**Table 4 ijms-24-16267-t004:** Availability of biofluids for protein biomarker analysis across the TBI disease spectrum.

Severity, Biofluids	Mild/Complicated Mild TBI (GCS: 13–15)	Moderate TBI (GCS: 9–12)	Severe TBI (GCS: 3–8)
Blood (plasma, serum) *	+	+	+
(Exosomes) **	+	NA	NA
Cerebrospinal fluid	-/+	+/-	+
Brain extracellular fluid (bECF; cerebral microdialysate) ***	-	-	+/-

**Abbreviations**: NA = not applicable; **Notes**: * More than 90% of protein biomarker studies—clinical and/or experimental—have used blood as biomaterial. ** Exosomes (extracellular or microvesicles) have gained much attention because of the potential to isolate them in an organ-specific manner and so their cargoes (miRNA and protein) can potentially offer the much-needed brain specificity. Their clinical utility in TBI is currently limited due to several unresolved theoretical and practical issues including factors affecting their release, life cycle, transport mechanism across membranes, etc., and, importantly, our currently limited knowledge about their biological functions and associations with pathobiological processes. From the technical point of view, the current isolation and analytical methodologies are not matured to the level of routine clinical use. *** Brain extracellular fluid (bECF) obtained with cerebral microdialysis (cMD) has not been routinely used.

**Table 7 ijms-24-16267-t007:** What Makes a Clinically Relevant Biofluid-Based Biomarker?

Criteria	Comments
Easily assayable in easily available biological fluid(s)	Typically, this will be blood (plasma/serum)
Standardized assay platform(s) available/does not rely on specialist technique for analysis	Assay platforms must be consistent across different hardware and software and over time if analyzed at multiple times; assays have to be robust, e.g., stability of the marker should be known within the biofluid if samples are not assayed immediately
High sensitivity and specificity	In order to supplement existing diagnostic modalities, such as head CT, high sensitivity and specificity are required
Rapid results	For clinical implementation, results must be available within a time frame that allows clinical decisions to be made; typically, this would be within an hour
Can be assayed multiple times	Timing in relation to injury has a major impact on interpretation
Reference ranges in health are available	Data on pathological thresholds and the range/standard deviation must be known in health and alternative pathologies with similar clinical presentations
Natural history or disease-related trajectory of response is well defined	Trajectory of change is more informative than one-off assessments
Clinicians have a clear conceptual understanding of what the result means	Ambiguous results or those that may be confounded by other factors (e.g., multi-system trauma, reduced hepatic or renal clearance) can cause additional problems for non-specialist decision makers
Result has a clear impact on management	Unless there is a specific fork in the clinical decision-making algorithm, additional information from a biomarker may not make a practical difference to the patient
Reduction in time in hospital/further investigations	‘Rule-out’ tests/triaging can be useful in speeding up patient flow within emergency departments, or avoiding unnecessary imaging
After a therapy is instituted	Given the paucity of potentially available therapies in TBI, implementation of biomarkers in clinical practice may be limited, but can indicate treatment efficacy

## Data Availability

Not applicable.

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
