# Peer review of "Fluid-Based Protein Biomarkers in Traumatic Brain Injury: The View from the Bedside"

_ijms, 2023, doi:10.3390/ijms242216267_

Round 1

Reviewer 1 Report

Comments and Suggestions for Authors

This manuscript has scientific merit. It covers a much needed topic in the field. The correlation between data from protein biomarkers and data of clinical diagnostic tests is much needed in the field of TBI. I think the authors have done an excellent effort in this respect and they have tried to fill this gap in TBI research. I have no issues concerning the scientific content of the manuscript, its readability should be enhanced. The language and structure of the manuscript needs to be enhanced to match the good content. I think that the authors should seek the help of native English speakers to sort out the many issues in the English language.  I suggest that the manuscript undergo major revision before acceptance.

Specific comments:

- Use a more concise title

- Use of references: Try to list each references next to the cited information. For example, lines 37-48 are all cited from references 4-6.   Also, here there is repettion of reference 26= “TBI where metabolic parameters are normalised while 66 alternative mechanisms but a potent inflammatory reaction may be at play [26, 27] [26, 28].”

- add references: The alternative approach to GCS-based classification is 71 to define biomarkers that relate not to ultimate functional outcome, and can thus be confounded by severity of injury, 72 but to specific ongoing pathological process(es). This accords much more closely with potential therapies that can be 73 translated from the animal literature 

- based on area under the ROC for TBI outcomes 35= Abstract is independent from manuscript. Define abbreviations upon first occurrence in abstract and in the rest of the manuscript= define ROC in introduction. Do the same for all abbreviations of the abstract and rest of manuscript.

- EXAMPLES of langauge corrections that should be made (These are only EXAMPLES): 

There has been  repeated at beginning of abstract.

of research into biofluid (blood, CSF) based= of research into biofluid-based (blood, CSF)

Partly driven by the aggregation of very large datasets such as those in CENTRE-TBI and TRACK-TBI, there is the potential now to correlate blood and CSF-based molecular (protein), radiological (imaging) and clinical (physiological) marker data to ultimate clinical outcomes= This is a long sentence and not clear= The presence of very large datasets, such as those in CENTRE-TBI and TRACK-TBI, allows for the correlation of blood- and CSF-based (protein), radiological (imaging), and clinical (physiological) marker data to advance clinical outcomes.

The quality of a given biomarker has often been framed in relation to 17 the predictive power on outcome= not clear

The ques- 20 tion thus arises as to how to incorporate important scientific results into patient management be- 21 cause, as of today, there is little or no standardized role for protein biomarkers in clinical decision 22 making.= long sentence= simplify.

based on area under the ROC for TBI outcomes 35= Abstract is independent from manuscript. Define abbreviations upon first occurrence in abstract and in the rest of the manuscript= define ROC in introduction.

based on area under the ROC for TBI outcomes= biomarker X or= a specific biomarker

In order to incorporate protein biomarker data into 37 current clinical practice so they can meaningfully contribute to clinical decision making= need proper punctuation= In order to incorporate protein biomarker data into 37 current clinical practice, so they can meaningfully contribute to clinical decision making,

TBI has been historically classified as Mild, Moderate and Severe [14-17] [18].= TBI has been historically classified as Mild, Moderate and Severe, according to the Glasgow Coma Scale (GCS) [14-17] [18].= use GCS only in the next sentence.

Conversely, there are examples of severe TBI where metabolic parameters are normalised while 66 alternative mechanisms but a potent inflammatory reaction may be at play= replace but with “such as   “ , for example.

Severe TBI, frequently co-morbid with polytrauma causes major loss of 80 brain parenchyma= add coma= Severe TBI, frequently co-morbid with polytrauma, causes major loss of 80 brain parenchyma

Line 117= The second dimension; = use colon= The second dimension:

Line 120= (see Figure).= which figure?= (Figure XX) with see.

Line 140-141: add another dash, remove repeated can= serially obtained biofluid-based mechanistic protein biomarker data can -in combination with data of neural damage- greatly help to identify the disease processes and their temporal pattern.

Comments on the Quality of English Language

Extensive English editing by a native speaker.

Author Response

                                                                                                Bethesda, November 07, 2023

Dear Reviewer,

We want to thank you for your thorough review.

Below please find our itemized, point-by-point response to your comments. Your critiques are copy pasted from your review, our responses are italicized. Revisions and additions are gray shaded in the revised manuscript.

We have thoroughly revised the manuscript following your recommendations and hope that the revised version is now acceptable for publication.

                        Sincerely,

Denes V. Agoston, M.D., Ph.D. and Adel Helmy, M.D.

Reviewer 1

This manuscript has scientific merit. It covers a much needed topic in the field. The correlation between data from protein biomarkers and data of clinical diagnostic tests is much needed in the field of TBI. I think the authors have done an excellent effort in this respect and they have tried to fill this gap in TBI research. I have no issues concerning the scientific content of the manuscript, its readability should be enhanced. The language and structure of the manuscript needs to be enhanced to match the good content. I think that the authors should seek the help of native English speakers to sort out the many issues in the English language.  I suggest that the manuscript undergo major revision before acceptance.

Specific comments:

- Use a more concise title

We have revised the tile per your recommendation and the new title reads: Fluid-based protein biomarkers in traumatic brain injury; the view from the bedside.

- Use of references: Try to list each references next to the cited information. For example, lines 37-48 are all cited from references 4-6.   Also, here there is repettion of reference 26= “TBI where metabolic parameters are normalised while 66 alternative mechanisms but a potent inflammatory reaction may be at play [26, 27] [26, 28].”

We thank you for pointing to these embarrassing oversights. We have corrected the references, eliminated duplications.

.

- add references: The alternative approach to GCS-based classification is 71 to define biomarkers that relate not to ultimate functional outcome, and can thus be confounded by severity of injury, 72 but to specific ongoing pathological process(es). This accords much more closely with potential therapies that can be 73 translated from the animal literature 

Corrected

- based on area under the ROC for TBI outcomes 35= Abstract is independent from manuscript. Define abbreviations upon first occurrence in abstract and in the rest of the manuscript= define ROC in introduction. Do the same for all abbreviations of the abstract and rest of manuscript.

Corrected

- EXAMPLES of langauge corrections that should be made (These are only EXAMPLES): 

There has been  repeated at beginning of abstract.

of research into biofluid (blood, CSF) based= of research into biofluid-based (blood, CSF)

Corrected

 Partly driven by the aggregation of very large datasets such as those in CENTRE-TBI and TRACK-TBI, there is the potential now to correlate blood and CSF-based molecular (protein), radiological (imaging) and clinical (physiological) marker data to ultimate clinical outcomes= This is a long sentence and not clear= The presence of very large datasets, such as those in CENTRE-TBI and TRACK-TBI, allows for the correlation of blood- and CSF-based (protein), radiological (imaging), and clinical (physiological) marker data to advance clinical outcomes.

Corrected

The quality of a given biomarker has often been framed in relation to 17 the predictive power on outcome= not clear

Sentence revised. It reads as: “Their predictive, or rather correlative power based on area under the Receiver Operating Characteristic (ROC) for TBI outcomes.”

The ques- 20 tion thus arises as to how to incorporate important scientific results into patient management be- 21 cause, as of today, there is little or no standardized role for protein biomarkers in clinical decision 22 making.= long sentence= simplify.

Corrected. It reads now as: “It is currently not known how to incorporate and to integrate biomarker data into patient management because there is no standardized role for protein biomarker data in clinical decision making.”

based on area under the ROC for TBI outcomes 35= Abstract is independent from manuscript. Define abbreviations upon first occurrence in abstract and in the rest of the manuscript= define ROC in introduction.

Corrected.

based on area under the ROC for TBI outcomes= biomarker X or= a specific biomarker

Corrected.

In order to incorporate protein biomarker data into 37 current clinical practice so they can meaningfully contribute to clinical decision making= need proper punctuation= In order to incorporate protein biomarker data into 37 current clinical practice, so they can meaningfully contribute to clinical decision making,

TBI has been historically classified as Mild, Moderate and Severe [14-17] [18].= TBI has been historically classified as Mild, Moderate and Severe, according to the Glasgow Coma Scale (GCS) [14-17] [18].= use GCS only in the next sentence.

Corrected.

-Conversely, there are examples of severe TBI where metabolic parameters are normalised while 66 alternative mechanisms but a potent inflammatory reaction may be at play= replace but with “such as   “ , for example.

Corrected.

-Severe TBI, frequently co-morbid with polytrauma causes major loss of 80 brain parenchyma= add coma= Severe TBI, frequently co-morbid with polytrauma, causes major loss of 80 brain parenchyma

Line 117= The second dimension; = use colon= The second dimension:

Line 120= (see Figure).= which figure?= (Figure XX) with see.

Corrected.

 Line 140-141: add another dash, remove repeated can= serially obtained biofluid-based mechanistic protein biomarker data can -in combination with data of neural damage- greatly help to identify the disease processes and their temporal pattern.

Corrected.

Comments on the Quality of English Language

Extensive English editing by a native speaker.

The revised manuscript has been edited by a native English speaker.

Reviewer 2 Report

Comments and Suggestions for Authors

The authors summarized the topic biomarker in patients with TBI as an extensive review. As the authors mentioned, the use of biomarker for clinical utility is not well established and there are lot of further research warranted to integrate the biomarkers in the current clinical practice. Further, it is important to integrate clinical, radiological as well as biomarker parameter to give an overall prognosis and this study gives us some hints to get into the step in the future. All in all, this is a well-written review and I have only minor issues to mention: 

-It would be very helpful to summarize the existing biomarkers with its sensitivity and specifity in a table format to have an overview of each biomarker. 
-Abstract: "There has been There has been" 
-Figure: Please define each colored graphic 

Author Response

                                                                                                            Bethesda, November 07, 2023

Dear Reviewer,                                                                                    

We want to thank you for your thorough review.

Below please find our itemized, point-by-point response to your comments. Your critiques are copy pasted from your review, our responses are italicized. Revisions and additions are gray shaded in the revised manuscript.

We have thoroughly revised the manuscript following your recommendations and hope that the revised version is now acceptable for publication.

                        Sincerely,

Denes V. Agoston, M.D., Ph.D. and Adel Helmy, M.D.

The authors summarized the topic biomarker in patients with TBI as an extensive review. As the authors mentioned, the use of biomarker for clinical utility is not well established and there are lot of further research warranted to integrate the biomarkers in the current clinical practice. Further, it is important to integrate clinical, radiological as well as biomarker parameter to give an overall prognosis and this study gives us some hints to get into the step in the future. All in all, this is a well-written review and I have only minor issues to mention: 

-It would be very helpful to summarize the existing biomarkers with its sensitivity and specifity in a table format to have an overview of each biomarker. 

A new table (Table 3) has been created listing the sensitivity and specificity of biomarkers.

-Abstract: "There has been There has been" 

Corrected, duplication removed.

-Figure: Please define each colored graphic 

Corrected.

Round 2

Reviewer 1 Report

Comments and Suggestions for Authors

Accept.

But, but care of these minor points:

Title= colon not semi-colon

Fluid-based protein biomarkers in traumatic brain injury; the view from the bedside= Fluid-based protein biomarkers in traumatic brain injury: the view from the bedside

In Abstract: 

biofluid- (blood, cerebrospinal fluid, CSF) based protein biomarkers in traumatic brain injury (TBI)= biofluid (blood, cerebrospinal fluid, CSF)-based protein biomarkers in traumatic brain injury (TBI).

blood-based protein biomarker data into patient management tool= blood-based protein biomarker data into patient management tools= or = blood-based protein biomarker data into a patient management tool

Comments on the Quality of English Language

English can still be enhanced.

Needs proofreading.